# Exploring Users’ Health Behavior Changes in Online Health Communities: Heuristic-Systematic Perspective Study

**DOI:** 10.3390/ijerph191811783

**Published:** 2022-09-18

**Authors:** Liyue Gong, Hao Jiang, Xusheng Wu, Yi Kong, Yunyun Gao, Hao Liu, Yi Guo, Dehua Hu

**Affiliations:** 1Department of Biomedical Information, School of Life Sciences, Central South University, Changsha 410013, China; 2Shenzhen Health Development Research and Data Management Center, Shenzhen 518028, China

**Keywords:** PADM, HSM, health behavior change, information processing, online health communities

## Abstract

(1) Background: With the continuous advancement of internet technology, use of the internet along with medical service provides a new solution to solve the shortage of medical resources and the uneven distribution of available resources. Online health communities (OHCs) that emerged at this historical moment have flourished with various advantages, such as being free from location and time constraints. Understanding users’ behavior changes via engagement in OHCs is necessary to support the development of internet medicine and promote public health. (2) Methods: The hypotheses of our research model were developed based on the protective action decision model (PADM) and heuristic-systematic model (HSM). A questionnaire was developed with seven constructs through previous studies and verified using a presurvey. Our survey respondents are online health community users. We used structural equation modelling to test the research hypotheses. (3) Results: The results of the analysis of 290 valid samples showed that the research model fit the data collected well. The perceived benefits (PB) positively affect information needs (IN) (beta = 0.280, *p* < 0.001, R^2^ = 0.252), thereby promoting users’ engagement in OHCs (EOHCs) (beta = 0.353, *p* < 0.001, R^2^ = 0.387); EOHCs has a significant positive impact on health behavior change (HBC) (beta = 0.314, *p* < 0.001), and it also significantly positively affects users’ health behavior change through systematic processing indirectly (beta = 0.252, *p* < 0.001, R^2^ = 0.387). (4) Conclusions: Our study offers support for the usefulness of the PADM and HSM in explaining users’ health behavior changes. For practitioners, this study introduces influence processes as policy tools that managers can employ for health-promoting with mHealth.

## 1. Introduction

With the rapid development and widespread use of internet technology, there are continuously growing online services and communication of all kinds occurring. Larger amounts of people are getting health information and using mobile health services through the internet. According to the Chinese Ministry of Industry and Information Technology, by the end of October 2021, the number of app downloads in the sports and health category was 500 billion [1].

With accelerated industrialization, urbanization, and population aging, Chinese residents’ lifestyle patterns and disease spectrum are constantly changing. Chronic non-communicable diseases such as cardiovascular diseases, cancer, chronic respiratory diseases, and diabetes account for 88% of all deaths and result in more than 70% of the total disease burden [2]. Low awareness rate of residents’ health knowledge and unhealthy lifestyles such as smoking, excessive drinking, lack of exercise, and unreasonable diets result in increasingly major disease problems. On October 18, 2017, President Xi Jinping pointed out in the report of the 19th National Congress that the Health China Strategy should be implemented [3]. Further, to achieve the goals of improving the national health policy and providing the people with a full range of whole-cycle health services, mobile health services are an effective means of implementing the policy.

OHCs are interactive online health platforms that integrate functions such as seeking medical advice, exchanging health information, getting access to health advice, social support, and emotional connection. It includes mobile health apps, health-related websites, and health forums. In China, Good Doctor Online and Sweet Home are two of the most typical OHCs. Different from physical medical services, OHCs exclude the limitations of time and space, offer the users easier and more economical access to information, and give space to some awkward health topics [4]. For people with chronic diseases, OHCs provide a platform for emotional connection and support [5]. OHCs have thrived on policy support in the past few years. Particularly under the context of the global pandemic, a reduction in going out and an increase in concern for information about the pandemic has led to a proliferation of mobile health services.

Most studies on OHCs focus on the evaluation of information service quality [6,7,8], users’ intention to use [9,10,11], and decision making [12]. Information processing is rarely covered, while user information processing is an essential direction in the study of OHCs. A small amount of research has focused on information processing approaches in OHCs, and the influence of information processing approaches on information credibility and adoption of opinions. For example, Sillence E collected data from four prostate patient communities and found that one-third of them used systematic processing in information decision making [13]. Another study used a heuristic-systematic processing model to explore the factors influencing the perceived credibility of diet and nutrition websites [14]. A study by Jin et al. proposed a healthcare information adoption model for online communities based on dual-process theory and knowledge adoption model [15]. However, these studies did not address the impact of information processing on users’ health behavior changes. OHCs can be used to promote healthy behavior and outcomes. A study suggest that the social influence (social integration, descriptive norms, and social support) exerted by online social relationships does affect the health behavior of users [16]. It has also been shown that information selection and acquisition can predict HPV vaccination intentions [17]. However, none of these studies discussed user’s health behavior changes in terms of the results of information processing. Our study explored the antecedents of information processing in OHCs and the results of information processing that produce individual behavior change.

One of the initial motivations for people to use OHCs is that they need accurate, valid, and adequate information to cope with the risk of disease in a situation where they are at risk of an illness or sub-health [18,19]. Moreover, participation in OHCs and information processing behavior triggered by the need for health information may also impact users’ attitudes towards health behavior change. To some extent, the impact of OHCs on people’s health behavior change exemplifies the role of OHCs in improving the population’s health literacy. Thus, exploring how OHCs promote users’ health behavior change has crucial practical implications. Some studies have examined the impact of information behavior on health behavior change in terms of information frameworks and demographic characteristics [20,21], as other studies suggest that internet interventions have a strong track record of promoting health behavior change [22]. However, few of them have paid attention to the information processing mechanism on health behavior change in OHCs. Based on the existing studies, this study attempts to investigate the influence of information processing on health behavior change in OHCs from the perspective of risk response to provide implications for the construction of OHCs and better persuade users to have healthier lifestyles.

Therefore, this study conducted in-depth research on the motivation of engagement in OHCs (i.e., perceived benefit, perceived risk, and information need) and information seeking based on the protective action decision model [23]. We further explored the effects of heuristic and systematic processing on HBC and determined which way has a more substantial impact on users’ HBC. Specifically, our research revolves around following questions: (1) What factors affect users’ engagement in OHCs? (2) How does information processing affect users’ health behavior change? (3) Which approach has a stronger effect on a users’ health behavior change?

This study will provide relevant authorities and administrators with e-health measures to guide population health behavior change and reveal the relationship between information processing and health behavior change.

## 2. Materials and Methods

### 2.1. Study Design and Participants

The aim of this study is to explore users’ health behavior changes in online health communities on the basis of the PADM and HSM. Ethical approval to conduct the study was obtained from the Institutional Review Board of College of Life Sciences, Central South University (Reference No:2021-1-44). WenJuanXing (https://www.wenjuan.com/, accessed on 8 March 2021) was used as a questionnaire editing and distribution tool. The questionnaire’s content consists of three parts. The first part is demographic characteristics. The information on the users’ usage of the OHCs was collected in the second part. The third part shows the measurement items for each latent variable of the research model. A total of eight latent variables included in the questionnaire are shown in Table 1, and a five-level Likert scale ranging from “strongly disagree” (1) to “strongly agree” (5) was adopted to quantify the observed items.

After completing the questionnaire design, the questionnaire was sent to 100 users of OHCs for a pre-survey in July 2021. We used exploratory factor analysis with varimax-rotated components to measure the validity of the designed questionnaire, and its cumulative total of variance and factor loadings were used to assess its construct validity. It was consistent with the constructs of the designed questionnaire if exogenous constructs accounted for more than 70% of the variance in the principal components. If each item had a factor loading value of 0.50 or higher on one of the principal components, but factor loading values below 0.50 on others, the validity of the designed questionnaire was considered acceptable. According to the result, “I’m not going to start trying to live a healthier lifestyle within six months” in the latent variable “Health Behavior Change” was removed. According to feedback and suggestions of respondents, some language adjustments and emphasis were made to make it easier and more accurate for the respondents to understand the meaning of the questionnaire.

Since the target population included users of OHCs, this survey distributed questionnaires online to platforms such as Sweet Home (https://bbs.tnbz.com/, accessed on 18 January 2021), a Chinese forum that provides diabetes information consultation and communication, health-themed Baidu postings, and Online patient communication groups to collect questionnaires. The sample size was determined by the general guidelines of structural equation modeling (SEM), with a recommended sample size of at least 10–15 times the number of scale items [31]. The collection period was from 13 July 2021, to 25 August 2021, and 303 questionnaires were received. A total of 290 valid questionnaires were obtained by deleting those with missing values, logical inconsistencies, and response times less than 90 s, with a sample recovery rate of 95.71%.

### 2.2. Analysis Strategy

Firstly, we used Cronbach alpha and composite reliability (CR) to measure reliability. Cronbach alpha coefficients were calculated using SPSS 26.0 for each latent variable. If the Cronbach alpha of each construct was 0.70 or higher, the reliability level of the scale was up to the mark [32]. SEM is a multivariate statistical technique for testing hypotheses about the influences of sets of variables on other variables [33]. It can measure the interrelation of latent variables that are not directly observable and is widely used in social sciences. We calculated the CR of each construct using SEM with the software program AMOS 23.0. Values of 0.7 or higher for CR for all constructs were considered acceptable [34]. To further examine the discriminant and convergent validity of the questionnaire, a confirmatory factor analysis was conducted to obtain values of the average variance extracted (AVE) and standardized loadings of items. The AVE of each construct and all of the standardized loadings should be greater than 0.50 [35]. Then, we tested the hypothesized relationships among the constructs by evaluating the structural model with the maximum likelihood method. Otherwise, one-way ANOVA was further used to assess the differences in health behavior change scores between the groups.

### 2.3. Research Hypothesis and Model Building

#### 2.3.1. Theoretical Background

(1)Protective Action Decision Model (PADM)

The protective action decision model describes the risk perceptions that individuals are triggered to consider in a risk situation to reduce risk by taking protective actions [23]. The process of protective action decision-making begins with environmental cues, social cues, and warnings. These perceptions combine situational facilitators and impediments to produce a behavioral response. However, the decision-making process may not include every stage in the protective behavioral decision model, nor will it necessarily be sequential. Some users engage in OHCs as information seeking, engagement, and processing behaviors triggered by disease risk. Individuals take these actions to help them cope with disease risk, leading to disease prevention, detection, treatment, and recovery. The use of mobile health services and information processing produces behavioral outcomes. This behavioral outcome manifests itself as a change in health behavior. This entire process is a protective behavioral decision process for the user. Based on this model, we further explored the result of decision implementation (i.e., the degree of health behavior change).

(2)Heuristic-Systematic Model (HSM)

PADM only emphasizes the vital role of information in behavior and does not consider the role of information processing in risk response and decision making [36]. This study thus explores users’ information behaviors in OHCs in conjunction with the heuristic-systematic model, which emphasizes the critical role of information processing in individual behavior, including systematic and heuristic information processing [29]. Systematic information processing refers to the tendency to use available information in a manner consistent with rationality by focusing on comprehensive analysis and stable judgment. In contrast, individuals use available intuition and experience during heuristic information processing to make decisions and expend less effort to gather more information [37].

#### 2.3.2. Perceived Risk, Perceived Benefit and Information Need

We set the factors related to user perception based on the PADM. Perceived risk is a critical factor for predicting people’s behavior in risk-taking situations [23]. Our view of health is that it is a reflection of the individual receiving signals about physical disease symptoms. When the individual is in a health-threatening situation, he needs sufficient health information to determine the disease status and cope with the disease’s risk. Uncertainty generates information needs [38], and the expectation that information needs will be met is the perceived benefit of their engagement in OHCs. There is a positive relationship between risk perception and information needs. The positive relationship between risk perception and information needs, in turn, influences subsequent information-seeking behavior [39].

OHCs provide users with a platform for health information consultation and experience sharing of similar cases. As a result, users engage in OHCs to seek adequate health information. Information seeking is a critical antecedent for users to use OHCs [18]. Therefore, the following hypotheses were proposed.

**H1:** *Users’ perceived risk positively influences their information needs*.

**H2:** *Users’ perceived benefit positively influences their information needs*.

**H3:** *Users’ information needs positively affect their engagement in OHCs*.

**H4:** *Users’ perceived risk positively affects their engagement in OHCs*.

**H5:** *Users’ perceived benefit positively affects their engagement in OHCs*.

#### 2.3.3. Engagement in OHCs and Health Behavior Change

User engagement is essential to the operation and development of OHCs, and it is partly a reflection of the success of OHCs [40]. In some kinds of research, user engagement is described as a psychological condition where the system is able to hold the user’s attention [41]. We treat specific manifestations of participation as a measure of user engagement, including user activity in OHCs and user interactions with each other. In previous studies, participation behavior is divided into reading (passive participation) and posting (active participation) [42]. Based on previous studies, user participation behavior in our study can be broadly divided into three categories: browsing behavior (search and browse information, but do not actively participate in discussions or post), interaction behavior (comments, likes, and retweets), and content creation behavior (post proactively).

More studies have explored the factors influencing engagement in OHCs, while fewer studies have focused on the outcomes of users’ engagement in OHCs [43]. Users’ active participation in OHCs has been shown to meet their information needs better; in this way, they feel a greater sense of belonging to the community [44]. The impact of this individual behavior change can be further explored. As a result of more health information and peer-to-peer support and encouragement, users with higher levels of participation may actively engage in information processing and affect health behavior changes [45]. User engagement enables them to create empowering and well-informed decisions about health care [46]. Thus, the following hypotheses were made.

**H6:** *Users’ engagement in OHCs affects their heuristic information processing*.

**H7:** *Users’ engagement in OHCs affects their systemic information processing*.

**H8:** *Users’ engagement in OHCs has a positive effect on their health behavior change*.

#### 2.3.4. Heuristic Information Processing and Systematic Information Processing

Information processing is an antecedent of attitude formation [47]. Interventions that guide behavior without trying to change beliefs and attitudes are often efficacious [48]. Therefore, information processing is one of the essential steps in promoting health behavior change. Beaudoin’s research suggests that health information obtained through mass media plays a crucial role in predicting health behaviors [49]. Plotnikoff’s study shows that the use of information resources that facilitate physical behavior can predict physical activity behavior in adults with type II diabetes [50].

Trumbo argued that SIP tends to make more stable judgments and subsequent behaviors than heuristic processing [29]. After a more systematic and comprehensive analysis of information, risks can be judged more clearly, and decisions made may be more motivated and valuable to implement as a result. Therefore, we established the following hypothesis:

**H9:** *Systematic information processing has a positive effect on health behavior change*.

**H10:** *Heuristic information processing has a positive impact on health behavior change*.

**H11:** *Systematic information processing has a greater degree of positive effects on health behavior change compared to heuristic processing*.

#### 2.3.5. Health Status and Health Behavior Change

From a risk response perspective, users with poorer health status are more likely to feel stressed [51], thus promoting health behavior change. In order to change this state of sub-health or unhealthy, there is a greater incentive to maintain healthy behaviors. One study has shown that people with poorer self-reported health prioritize their health [52], reflecting that people in worse health status are more aware of their health. This attention may result in health behavior changes. Based on this, we constructed the following hypothesis.

**H12:** *Health status is negatively associated with health behavior change*.

#### 2.3.6. Model Building

We constructed a structural equation model based on the PADM and HSM (Figure 1). Since H11 was based on the comparison of path coefficients of H9 and H10, and H12 was validated by a one-way ANOVA, H11 and H12 were not included in the structural equation model. This process reflects users’ protective behavioral decision-making process in OHCs and its outcomes.

## 3. Results

### 3.1. Sample Characteristics

The demographics of the respondents are shown in Table 2. The number of women was slightly higher than men. About half of the education degrees were bachelor’s degrees, and the least percentage of master’s degrees and above was 6.90%. The income ranged from 5000 yuan to 10,000 yuan. To survey respondents clearly understand the frequency of OHC use, we limited the frequency of use to the last month. The frequency of OHCs was the largest group of people “occasionally” and “sometimes,” 27.59% and 37.59%, respectively. Furthermore, users are more likely to use OHCs when in a stressful health condition. The majority of participants were between 20 and 30 years of age, accounting for 56.21%.

One-way ANOVA suggested that there are statistically significant differences in health behavior change between groups with respect to education level, income, frequency of using the OHC, and health status. Users with higher education, income and more frequent use of online health communities scored higher on health behavior change, while users with poorer health status showed greater health behavior change (*p* < 0.001). Therefore, H12 was tested.

### 3.2. Measurement Model Testing

Table 3 shows the reliability of the questionnaire; the standardized factor loadings of each latent variable were greater than 0.50, indicating that each measurement question item can reflect the latent variables better. The AVE ranges between 0.550 and 0.670, which was higher than 0.50. Values of CR were between 0.787 and 0.859, which were higher than 0.70, indicating that constructs had good convergent validity. The factor loadings of each construct were much greater than the cross-loadings on other constructs, and correlations of the constructs were much smaller than the square root of the average variance extracted from each construct, indicating discriminant validity [53], as shown in Table 4.

The model-fit-indices were as follows: χ2/df = 1.859, CFI (Comparative Fit Index) = 0.944, GFI (Goodness-of-Fit Index) = 0.904, TLI (Tucker–Lewis Index) = 0.932, and RMSEA (Root-Mean-Square Error of Approximation) = 0.055, which indicates that the research model fitted the collected data well (CFI > 0.9, GFI > 0.9, TLI > 0.9, RMSEA < 0.05). All fit indexes were within the range of the recommended values, indicating that the research model fit the data collected well.

### 3.3. Structural Model Testing

To exclude confounding effects in demographic variables, we included variables which have statistically significant differences in user health behavior change in our research model as control variables. The results of testing all path coefficients are shown in Table 5. Both perceived risk and perceived benefit positively affect users’ information needs. Users’ information needs significantly affects their engagement in OHCs. Hypothesis H1–3 was supported. The effect of perceived risk on engagement in OHCs was not significant at a *p*-value 0.05 level. H4 was not supported. Users’ perceived benefit significantly affects their engagement in OHCs (beta = 0.422, *p <* 0.0001), and H5 was supported. H6 was supported for EOHCs having a negative effect on HIP (beta= −0.149, *p* = 0.037). Users’ engagement in OHCs positively impacts both SIP and health behavior change (*p <* 0.001). Therefore, H7 and H8 were supported. H9 was confirmed for the positive effect of SIP on health behavior change, while the effect of HIP on health behavior change was not verified, H10 was not supported; since the result of heuristic processing on health behavior change was not statistically significant, while systematic processing has a significant effect on it, H11 was supported. In summary, two of the eleven hypotheses in our study were not supported, and the rest were kept (as shown in Figure 2).

## 4. Discussion

### 4.1. Principal Findings

Our research found that the antecedents of EOHCs were information needs and perceived benefits, and that EOHCs and active SIP were positively related to changes in OHCs user behavior. Users who were more actively engaged in OHCs employed less HIP. The perceived risk of overcoming disease threats triggers the information needs of OHCs users, which leads to their engagement in OHCs. In response to signals about their physical condition, people often need more accurate health information to counter disease threats or stay healthy. Hang Lu et al. found a positive impact of risk response on the increasing information need and information engagement intentions to reduce uncertainty [54]. Their results are somewhat similar to our study. A health belief model suggests that people will adopt a behavior if they perceive that there are more benefits than barriers associated with it [55]. Individuals may consider how changes in their lives can have a positive effect on behaviors that reduce illness risk [38]. This conclusion is partially supported by our findings. An important factor driving engagement in OHCs is perceived benefits.

It is noteworthy that the effect of perceived risk on EOHCs does not hold. This may own to the reason that users with excessive perceived risk will seek help directly from offline hospitals due to the unavailability of direct diagnostic advice and medical equipment resources online at the threat of serious illness, and they do not get deeper involved in OHCs. From another perspective, lower perceived risk may indicate that users are less worried about their health status and therefore less motivated to participate in OHCs.

Additionally, we found that EOHCs positively influenced users’ health behavior. This effect is particularly pronounced under conditions of systematic information processing. Having a healthy lifestyle is one of the most important means of disease prevention [53,56]. OHCs play an important role in regulating and promoting healthy living among residents, while the way users process information affects their persuasive effect [33]. The higher the level of participation, the greater the sense of belonging to a community, creating a positive feedback loop [57,58]. The more carefully and systematically the users process the information, the more significant the effect on health behavior change of systematic information processing.

Conversely, a comprehensive and systematic approach to information processing requires that users be more engaged in OHCs. There are fewer contexts in which HIP is likely to be adopted if users engage more in OHCs. This explains our finding: EOHCs has a negative effect on heuristic information processing. In addition, it is possible that the endorsement preference for different information processing methods is related to the user’s personality and habits; people used to HIP are more inclined to irrational thinking and may be more uncomfortable with the constraints of rules [59], and thus less likely to develop healthier behaviors. Thus, the positive effect of HIP on health behavior change has not been verified.

In addition to the above findings, we also found that income and education level affect the degree of health behavior change. This is consistent with the results of a previous study: specific health behavior correlates with a specific level of education and socio-economic status [60,61]. Part of the reason for this is the fact that education level and economic status are generally correlated. In our study, users with higher income, and with master’s degrees and above have a stronger intention to change health behavior. The frequency of OHCs use, the greater the change in users’ health behaviors. Frequent use of OHCs may result from users’ anxiety about their health conditions; changes in health-related behavior are closely related to health status as well as fears and wishes regarding health [60], which is also in accordance with our finding that users with worse health status had more health behavior changes.

### 4.2. Implications

#### 4.2.1. Theoretical Implications

First, our study applied PADM and HSM in health behavior change to explore the correlation between users’ information processing in OHCs and their health behavior change. The connection between cognitive processes and users’ health behavior provides a new application direction for information processing theory. Secondly, our research focuses on the results of users’ participation in OHCs and their information processing, which broadened the scope of research in OHCs. Finally, our study also explores the antecedents of information processing from a risk response perspective, explaining the motivation of users to participate in OHCs and information processing. The findings can inform future research.

#### 4.2.2. Practical Implications

Our study showed that engagement in OHCs can promote users’ health behaviors, and therefore relevant health authorities can conduct interventions to promote population health through OHCs. Based on our results, relevant public health providers should take measures to better meet users’ information needs in terms of perceived risks and perceived benefits to increase their participation in OHCs. For example, developing a health status self-assessment function would allow users to be aware of their health status. But again, care should be taken to ensure that the test is fair, as excessive perceptions of risk may lead users to turn directly to offline counseling. Managers can also enhance users’ perceived benefits by successfully pushing information about users’ successful cases regarding health behavior change, thus attracting them to participate more in OHCs. As users are motivated to increase their level of engagement, they will engage in systematic information processing and healthily change their behavior. Secondly, users can be motivated to systematically process and promote healthy life changes by presenting more systematic information, for example, by recommending content related to users’ search terms so that users can have a more comprehensive and systematic understanding of relevant health information. Besides, for users, if they want to maintain health, they need to keep a healthy lifestyle or make good change in health behavior. More active participation in OHCs and the adoption of systematic information processing can make health behavior changes more effective.

### 4.3. Limitations and Directions for Future Research

This study has several limitations. Firstly, cross-sectional studies are considered to be incapable of causal inference due to synchronicity of covariate surveys, statistical associations, and survivorship bias. Longitudinal surveys may be considered in future studies to remedy this deficiency [62]. Secondly, using questionnaires to obtain data on users’ information processing is too subjective. Other objective methods such as eye-tracking could be used in future studies to quantify users’ information processing. Thirdly, there is no breakdown of users’ health behavior change into health behavior categories, and future studies could further explore the effect of information processing on a specific health behavior change and the extent of different categories of health behaviors. Fourthly, due to the impact of the COVID-19 pandemic, we used an online survey. This approach may be less appropriate for older adults, with a smaller sample of older adults obtained in the results, and thus our results may be biased by age distribution.

## 5. Conclusions

This study explored the effects of user engagement and information processing on their health behavior change in OHCs. Our findings provide empirical support for the important role that OHCs play in promoting the health of their users, which provides health service workers with a reliable and explorable approach for health interventions. Specifically, perceived benefits and perceived risks increase users’ information needs, influencing their degree of engagement in OHCs. Users more engaged in OHCs were more likely to use systematic information processing, thus changing their health behavior. However, since our sample size did not include a sufficient number of older adult groups, the applicability of the study findings to older adult groups needs to be further explored. It is worth noting that older adults or those with chronic conditions are more likely to benefit from OHCs, so further exploration of how this group benefits from online health communities is an essential direction for research. Administrators and policymakers must also be aware of the unfamiliarity of older adults with OHCs and design more accessible and user-friendly ways to promote their use of OHCs. According to our findings, managers of OHCs and relevant policymakers can take appropriate measures to promote residents’ health behavior change through OHCs.

## Figures and Tables

**Figure 1 ijerph-19-11783-f001:**
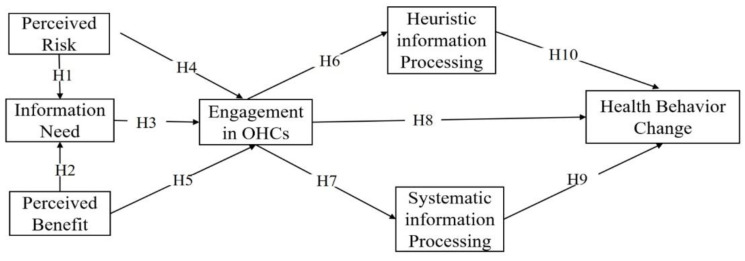
The conceptual research model based on PADM and HSM. H1–H10: Hypothesis 1–10.

**Figure 2 ijerph-19-11783-f002:**
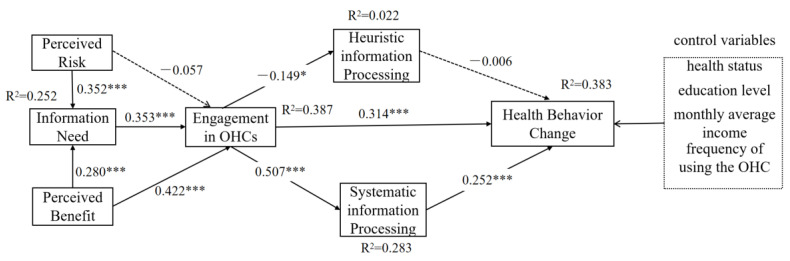
The conceptual research model based on PADM and HSM and the results of the maximum likelihood estimate. * *p* < 0.05; *** *p* < 0.001.

**Table 1 ijerph-19-11783-t001:** Measurement items of the constructs.

Constructs	Measurement Items	Source
Perceived Risk (PR)	How worried are you about your health condition?	Leiserowitz, A. [24]
	How serious of a threat do you believe disease is to health?	
	How serious are the current impacts of disease?	
Perceived Benefit (PB)	Health information can be of benefit to me in managing my health.	Bhattacherjee, A. [25]
	Having more health information can help me deal with disease threats.	
	Health information can enhance my effectiveness in managing my health.	
Information Need (IN)	I want to seek health information.	E. Ter Huurne and J. Gutteling [26]
	I have to seek more health information.	
	I follow health information through multiple channels.	
Engagement in OHCs (EOHCs)	I browse the information in the online health community.	
	I like, comment, reply or retweet other users’ posts in the online health community.	
	I post original posts in the online health community.	
Heuristic Information Processing (HIP)	I do not evaluate the quality of health information when using the online health community.	Smerecnik, CMR et al. [27] Wong, JCS et al. [28]
	When using the online health community, I don’t think twice about adopting the health information I find.	
	I won’t spend much time thinking about health information when I use the online health community.	
Systematic Information Processing (SIP)	When using the online health community, I thought about how the information related to other things I know.	Trumbo, CW [29]
	When using the online health community, I found myself making connections between the information and what I’ve read or heard about elsewhere.	
	When using the online health community, I tried to relate the ideas in the information to my health.	
Health Behavior Change (HBC)	I’m not going to start trying to live a healthier lifestyle within 6 months.	Wang CL et al. [30]
	In six months, I plan to start trying to live a healthier lifestyle.	
	I’m going to start trying to live a healthier lifestyle in 30 days.	
	I have only started a healthier lifestyle within the last 6 months.	
	I am maintaining a healthier lifestyle and have been for more than 6 months.	

**Table 2 ijerph-19-11783-t002:** Demographic characteristics of the participants (N = 290).

Variables	Measure and Category	Value (N = 255), n (%)	*p*-Value ^1^
Sex			
	Male	137(47.24)	0.257
	Female	153(52.76)	
Age group			
	20–30	163(53.21)	
	31–40	125(43.10)	0.094
	41–50	2(0.69)	
Education level			
	Up to secondary school	56(19.31)	
	Junior College	70(24.14)	0.048 *
	Undergraduate	144(49.66)	
	Postgraduate and higher	20(6.90)	
Monthly average income (yuan)			
	<5000	48(16.55)	
	5000–10,000	155(53.45)	0.039 *
	>10,000	87(30.00)	
Frequency of using the OHCs			
	Occasionally	80(27.59)	
	Sometimes	109(37.59)	***
	Frequently	62(21.38)	
	Always	19(6.55)	
Medical Background			
	Yes	58(20.00)	0.449
	No	232(80.00)	
Health Status			
	Very bad	11(3.79)	
	Bad	42(14.48)	
	General	72(24.83)	***
	Good	98(33.79)	
	Very good	67(23.10)	

^1^ *: *p*-Value < 0.05; ***: *p*-Value < 0.001.

**Table 3 ijerph-19-11783-t003:** Statistical results of the research model.

Constructs	Items	Standard Loadings	Cronbach Alpha	AVE ^1^	CR ^2^
PR	PR1	0.760	0.833	0.628	0.835
	PR2	0.778			
	PR3	0.838			
PB	PB1	0.742	0.821	0.610	0.824
	PB2	0.839			
	PB3	0.758			
IN	IN1	0.797	0.854	0.670	0.859
	IN2	0.893			
	IN3	0.760			
EOHCs	EO1	0.744	0.786	0.552	0.787
	EO2	0.747			
	EO3	0.738			
HIP	HIP1	0.778	0.829	0.620	0.830
	HIP2	0.785			
	HIP3	0.799			
SIP	SIP1	0.799	0.816	0.595	0.815
	SIP2	0.784			
	SIP3	0.730			
HBC	HBC1	0.680	0.831	0.550	0.830
	HBC2	0.754			
	HBC3	0.733			
	HBC4	0.795			

^1^ AVE is the average variance extracted from the model; ^2^ CR is composite reliability.

**Table 4 ijerph-19-11783-t004:** Correlation matrix (N = 290).

	PR	PB	IN	EOHCs	SIP	HIP	HBC
PR	0.792 ^1^						
PB	0.327	0.781					
IN	0.421	0.414	0.819				
EOHCs	0.318	0.555	0.560	0.743			
SIP	0.198	0.345	0.349	0.622	0.771		
HIP	0.046	0.081	0.082	0.146	0.091	0.787	
HBC	0.198	0.346	0.349	0.622	0.578	0.111	0.742

^1^ The value of the diagonal is the square root of the average variance extracted from each construct.

**Table 5 ijerph-19-11783-t005:** Hypothesis testing results of the research model.

Hypothesis Paths			Unstandardized Path Coefficients	Standardized Path Coefficients	*p*-Value	Results
PR	→	IN	0.342	0.352	***	H1 supported
PB	→	IN	0.260	0.280	***	H2 supported
IN	→	EOHCs	0.343	0.353	***	H3 supported
PR	→	EOHCs	−0.053	−0.057	0.434	H4 not supported
PB	→	EOHCs	0.381	0.422	***	H5 supported
EOHCs	→	HIP	−0.163	−0.149	0.037 *	H6 supported
EOHCs	→	SIP	0.526	0.507	***	H7 supported
EOHCs	→	HBC	0.281	0.314	***	H8 supported
SIP	→	HBC	0.217	0.252	0.002 ***	H9 supported
HIP	→	HBC	−0.005	−0.006	0.927	H10 not supported

*: *p*-Value < 0.05; ***: *p*-Value < 0.001.

## Data Availability

The data that support the findings of this study are available from the corresponding author upon reasonable request.

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
