# Peer review of "Exploring Users’ Health Behavior Changes in Online Health Communities: Heuristic-Systematic Perspective Study"

_ijerph, 2022, doi:10.3390/ijerph191811783_

Round 1
Reviewer 1 Report
1. Introduction
The relevant research introduces the research on different aspects of OHC, and should increase the relevant research on OHC users' health behavior, focusing on analyzinghow does your study differ from other studies of OHC users' health behavior.( Line61-75)
4. Discussion
4.1. Principal Findings
The discussion section of this study resembles the results section. In general, hypothesis verification based on the analysis results is performed in the result part. In addition, the discussion section should not only summarize the results but also produce noticeable (theoretical and practical) implications according to the results. The present discussion lacks implications.(Line309-356)
Author Response
Dear Reviewer:
All authors would like to thank you for your time and efforts spent on revising our manuscript. Your detailed and constructive comments are of significance for us to improve the quality of our research. We have read your comments carefully and each comment will be addressed regarding the modified manuscript with “Track Changes” .
Point 1: The relevant research introduces the research on different aspects of OHC, and should increase the relevant research on OHC users' health behavior, focusing on analyzing how does your study differ from other studies of OHC users' health behavior.( Line61-75)
Response 1: Thank you very much for your comments and professional advice. We added the relevant research on OHC users' health behavior(Page2, Line73-78). And the difference between our study and previous study is that our study address the impact of information processing on users' health behavior changes in OHCs,we emphasized the difference in page2, line71-72, 77-80,91-96.
Point 2: The discussion section of this study resembles the results section. In general, hypothesis verification based on the analysis results is performed in the result part. In addition, the discussion section should not only summarize the results but also produce noticeable (theoretical and practical) implications according to the results. The present discussion lacks implications.(Line309-356)
Response 2: The principal finding in the Discussion section is a discussion of the results, including comparison with results from other literature and explaining why the hypothesis did not hold. The purpose of this section is to further explore the implications of the study. So we think this part is justified in the Discussion section. Based on your suggestion, we have added theoretical and practical implications in the Discussion section (page 11,line 370-387).

Reviewer 2 Report
This manuscript presented a user study to understand the users' behavior change via engagement in online health communities (OHCs). Authors gave a clear presentation. Especially, both the hypotheses and questions were well defined. Results are well explained and presented with an appropriate statistic method.
However, the study is limited to young participants (ages between 20-40), while elder people are more likely to suffer from chronic disease and may benefit more from online health communities. It is not clear whether the results can be generalized to elder users.
In the result, authors indicated that there are statistically significant differences in health behavior change between groups with respect to health status. It is interesting to know how the health status affects the behavior change and explain the difference. The majority of the participants are healthy people (general or above). Can the results also applied to people with chronic diseases?
I would recommend to expand the discussion on principle findings. For example, how the findings can be applied to improve the engagement in the existing online health communities?
In summary, this manuscript is well written. However, my major concern is the limitation on the sample population, which did not cover elder people with poor health status. It raises the generalizability issue.
Author Response
Dear Reviewer:
All authors would like to thank you for your time and efforts spent on revising our manuscript. Your detailed and constructive comments are of significance for us to improve the quality of our research. We have read your comments carefully and each comment will be addressed regarding the modified manuscript with “Track Changes” .
Point 1: However, the study is limited to young participants (ages between 20-40), while elder people are more likely to suffer from chronic disease and may benefit more from online health communities. It is not clear whether the results can be generalized to elder users.
Response 1: Our study was conducted during the time of the COVID-19 pandemic, and in accordance with Chinese epidemic prevention policies (e.g., keeping social distance and reducing gatherings), we were limited to online surveys. Due to unfamiliarity with the Internet, older adults need active guidance and encouragement to fill out questionnaires, which is hard to do with online surveys, making it difficult to receive responses from older adults for our survey. We have elaborated this limitation in the discussion section (page11, line 397-400). The use of online health communities by the group of elder adults and how they benefit from OHCs can be further explored in future studies.
Point 2: In the result, authors indicated that there are statistically significant differences in health behavior change between groups with respect to health status. It is interesting to know how the health status affects the behavior change and explain the difference. The majority of the participants are healthy people (general or above). Can the results also applied to people with chronic diseases?
Response 2: Because our study did not aim to explore the impact of demographic characteristic factors on users of online health communities, we did not focus on the influence of health status on behavior change in the Results section. We do, however, mention this in the principal findings in the Discussion section (page11, line 359-369). From a risk response perspective, users with poorer health status are more likely to feel stressed, thus promoting health behavior change. In order to change this state of sub-health or unhealthy, there is a greater incentive to maintain healthy behaviors. One study have shown that people with poorer self-reported health prioritize their health, reflecting that people in worse health status are more aware of their health, resulting in health behavior change. The relevant literature is listed below this response. To protect user privacy, we do not investigate users' diseases, but the online health communities we investigated include the diabetes community, which is focused on exchanging experiences, so we think this result can be used to some extent for people with chronic diseases.
Reference:
- Marzec ML, Lee SP, Cornwell TB, Burton WN, McMullen J, Edington DW. Predictors of behavior change intention using health risk appraisal data. Am J Health Behav. 2013 Jul;37(4):478-90. doi: 10.5993/AJHB.37.4.6. PMID: 23985229.
- Meyerhof H, Jones CM, Schüz B. Intra-individual trajectories of subjectively prioritizing health over other life domains. Appl Psychol Health Well Being. 2022 May 17. doi: 10.1111/aphw.12368. Epub ahead of print. PMID: 35578834.
Point 3: I would recommend to expand the discussion on principle findings. For example, how the findings can be applied to improve the engagement in the existing online health communities?
Response 3: Thank you very much for your comments and professional advice. We have added theoretical and practical implications in the Discussion section (page 11,line 370-387).

Round 2
Reviewer 2 Report
Thank you for revising the manuscript. However, my concerns are not completely addressed in the revision.
Point 1: In-person interview may not be feasible during pandemic, but it is possible to conduct online surveys. Especially, the pandemic is already easing. At least, authors should revise the manuscript throughout, revising the conclusion based on the sample population.
Point 2: Health status seems a fundamental reason that affects user behavior. Health status is co-related to risk, information and benefit. Should health status a hypothesis being tested?
Point 3: The description in 4.2.2. is very general and do not provide a direct connection between results and those practical implications. For example, "choosing more reputable doctors and improving the credibility of information" is just common sense. Even without doing a survey, we can still get the above conclusion.
Author Response
Dear Reviewer:
All authors would like to thank you for your time and efforts spent on revising our manuscript. Your detailed and constructive comments are of significance for us to improve the quality of our research. We have read your comments carefully and each comment will be addressed regarding the modified manuscript with “Track Changes” .
Point 1: In-person interview may not be feasible during pandemic, but it is possible to conduct online surveys. Especially, the pandemic is already easing. At least, authors should revise the manuscript throughout, revising the conclusion based on the sample population.
Response 1:
We are sorry to that our study covered a relatively small group of older adults. We recruited participants through OHCs in China(e.g., Sweet Home, Good Doctor Online), and unfortunately received few responses from older adults. The possible reasons for this are, firstly, as the unfamiliarity of the elderly group with using the Internet, older users represent a smaller percentage of overall OHC users; The second is that older adults may be less willing to participate in surveys. This is indeed one of the limitations of the study. Based on your suggestion, we have revised the conclusion section based on the sample population, noting that future studies could target the older population or those with chronic conditions and explore how they benefit from OHCs and how to increase their participation(page12,line427-442).
Point 2: Health status seems a fundamental reason that affects user behavior. Health status is co-related to risk, information and benefit. Should health status a hypothesis being tested?
Response 2:
Our study focused on constructing a structural equation model based on the purpose of the study (to explore the effects of engagement and information processing in online health communities on health behavior change) and theoretical foundations (HSM and PADM).We designed the questionnaire based on the selection of variables in the structural equation model. Structural equation modeling mainly explores the interrelationships between latent variables. Since health status is not a latent variable but a categorical variable in our survey, incorporating the hypothesized path of health status on other variables in the structural equation model is not the most appropriate statistical approach. Therefore, in the demographic characteristics section, we explored the effect of health status on the outcome variable (health behavior change) using univariate analysis and included it as a control variable in the structural equation model.
Based on your suggestion, we added hypothesis 12. Although we cannot count the importance of health status to the path hypothesis of the structural equation model, we can test the hypothesis by means of univariate analysis and elaborate again in the results section on the differences in health behavior change for groups with different health status(page6,line252-258; page7,line279-284).
Point 3: The description in 4.2.2. is very general and do not provide a direct connection between results and those practical implications. For example, "choosing more reputable doctors and improving the credibility of information" is just common sense. Even without doing a survey, we can still get the above conclusion.
Response 3: Thank you very much for your pertinent suggestions. According to your advices, we have proposed more detailed implications based on the results(page11,line393-411).